# Charge Order and Suppression of Superconductivity in HgBa$_2$CuO$_{4+d}$ at High Pressures

Manuel Izquierdo [1,2], Daniele C. Freitas [3,4], Dorothée Colson [5], Gastón Garbarino [6], Anne Forget [5], Helène Raffy [7], Jean-Paul Itié [2], Sylvain Ravy [2,7], Pierre Fertey [2] and Manuel Núñez-Regueiro [3,*]

1   European XFEL GmbH, Albert Einstein Ring 19, 22761 Hamburg, Germany; manuel.izquierdo@xfel.eu
2   Synchrotron SOLEIL, CEDEX, 91192 Gif-sur-Yvette, France; jean-paul.itie@synchrotron-soleil.fr (J.-P.I.); sylvain.ravy@u-psud.fr (S.R.); pierre.fertey@synchrotron-soleil.fr (P.F.)
3   Centro Brasileiro de Pesquisas Fisicas, Rua Dr. Xavier Sigaud, 150, Urca, Rio de Janeiro 22290-180, RJ, Brazil; danielecsf@id.uff.br
4   Institut Néel, Université Grenoble Alpes, Centre National de la Recherche Scientifique, 25 rue des Martyrs, BP 166, CEDEX 9, 38042 Grenoble, France
5   CEA, Université Paris-Saclay, Centre National de la Recherche Scientifique, SPEC, 91191 Gif-sur-Yvette, France; dorothee.colson@cea.fr (D.C.); anne.forget@cea.fr (A.F.)
6   European Synchrotron Radiation Facility (ESRF), BP 220, CEDEX 9, 38043 Grenoble, France; gaston.garbarino@esrf.fr
7   Laboratoire de Physique des Solides, Université Paris-Sud 11, Centre National de la Recherche Scientifique, CEDEX, 91405 Orsay, France; raffy@lps.u-psud.fr
*   Correspondence: nunez@neel.cnrs.fr

**Abstract:** New insight into the superconducting properties of *HgBa$_2$CuO$_4$* (*Hg-1201*) cuprates is provided by combined measurements of electrical resistivity and single crystal X-ray diffraction under pressure. The changes induced by increasing pressure up to 20 GPa in optimally doped single crystals were investigated. The resistivity measurements as a function of temperature show a metallic behavior up to ~10 GPa that gradually passes into an insulating state, typical of charge ordering, which totally suppresses superconductivity above 13 GPa. The changes in resistivity are accompanied by the apparition of sharp Bragg peaks in the X-ray diffraction patterns, indicating that the charge ordering is accompanied by a 3D oxygen ordering. Considering that pressure induces a charge transfer of about 0.02 at 10 GPa, our results are the first observation of charge order competing with superconductivity developed in the overdoped region of the phase diagram of a Hg-based cuprate.

**Keywords:** superconducting cuprates; high pressure; electrical resistivity; X-ray crystallography; charge density waves

## 1. Introduction

In recent years, the observation of charge order (CO) [1–3] in cuprates other than the well-studied one [4] of *La$_{2-x}$Ba$_x$CuO$_4$* has led to a boom of studies in the hope of finding the key to understanding high-temperature superconductors. Of particular interest are the charge density wave (CDW) fluctuations observed in *Hg-1201* [5,6]. In opposition to other families, the mercury family has flat tetragonal *CuO$_2$* planes, and there is apparently no plane distortion but an oxygen ordering that determines the charge density wave. Most interestingly, the (H) component of the CDW wave vector has been shown to scale [6] with those determined for the *YBa$_2$Cu$_3$O$_{7-x}$* (*YBCO)* system as a function of doping. Diffraction studies have shown that oxygen ordering has a one-dimensional character that manifests as diffuse lines due to the fluctuating charge ordering in the two tetragonal directions [7,8]. More recently, a phase separation scenario has been proposed by scanning micro-X-ray diffraction, in which CDW regions and oxygen interstitial regions coexist [9].

In order to understand the relevance of oxygen ordering in the superconductivity of *Hg-1201*, which exhibits only slight intensity modifications upon changing temperature [7],

we decided to perform experiments under pressure. This variable has been shown to increase the transition temperature in cuprates, particularly in the case of mercury compounds. Thus, the highest superconducting transition temperature ($T_c$) reported to date, $T_c$ = 166 K, was on *Hg1223-F* at 25 GPa [10]. The mechanisms controlling this increase have been thoroughly studied [11–14]. Normally, the leading mechanism is the charge transfer under pressure, due to the strong compression along the *c* axis, which reduces the ionicity of the layers and causes a passage of electrons from the negatively charged $CuO_2$ layers to the reservoir layers. The result is a parabolic variation of $T_c$ under pressure. The compression of the *a* parameter, involving a significant shortening of the *CuO* bond [15], can induce a strong linear variation of $T_c$ which has been used to explain the anomalously strong increase observed in the flat $CuO_2$ plane *Hg* cuprates [16]. It should also affect the fluctuating one-dimensional oxygen ordering recently reported. Furthermore, a correlation between $T_c$ and the changes in oxygen ordering are to be expected when the latter is relevant for superconductivity. This relation has indeed been demonstrated to occur in other cuprate families [17].

## 2. Experimental

We performed electrical resistivity and single-crystal X-ray diffraction studies under pressure on an optimally doped *Hg-1201*. Single crystals were synthesized using a flux technique by identifying the most favorable region of the ternary diagram *HgO-BaO-CuO* to obtain *Hg-1201* single crystals. They have well-developed (001) faces with very clean surfaces and a size in the range of $0.3 \times 0.3 \times 0.3$ mm$^3$. The critical temperature, measured with a Superconducting Quantum Interference Device (SQUID) magnetometer, showed a transition onset at $T_c$ = 95 K and a narrow width (~4 K) for isolated single crystals, thus confirming a high sample quality [18].

## 3. Results

### 3.1. Electrical Resistivity

Electrical resistivity was measured in a solid-state pressure cell. In Figure 1b, the electrical resistance of a *Hg-1201* single crystal as a function of temperature and pressure is displayed. At the lowest pressure, the behavior is clearly metallic with a sharp superconducting transition. As pressure increases, the electrical resistivity decreases up to about 5 GPa. At higher pressures, the resistivity starts increasing and the superconducting transition widens. The last faint signature of a superconducting transition is observable at 11.5 GPa. At higher pressures, the sample shows an activated behavior typical of an insulator. This can be due to either some sort of pressure-induced ordering or sample degradation. Even though the solid-state pressure cell is not conceived to measure on decompression, we performed a measurement at 4 GPa on decompression. We observed that the sample recovered the superconducting state, but not the metallic character. This can be due to sample degradation at high pressures, to the non-homogenous strains due to the decompression of the solid-state cell, or both Figure 1c.

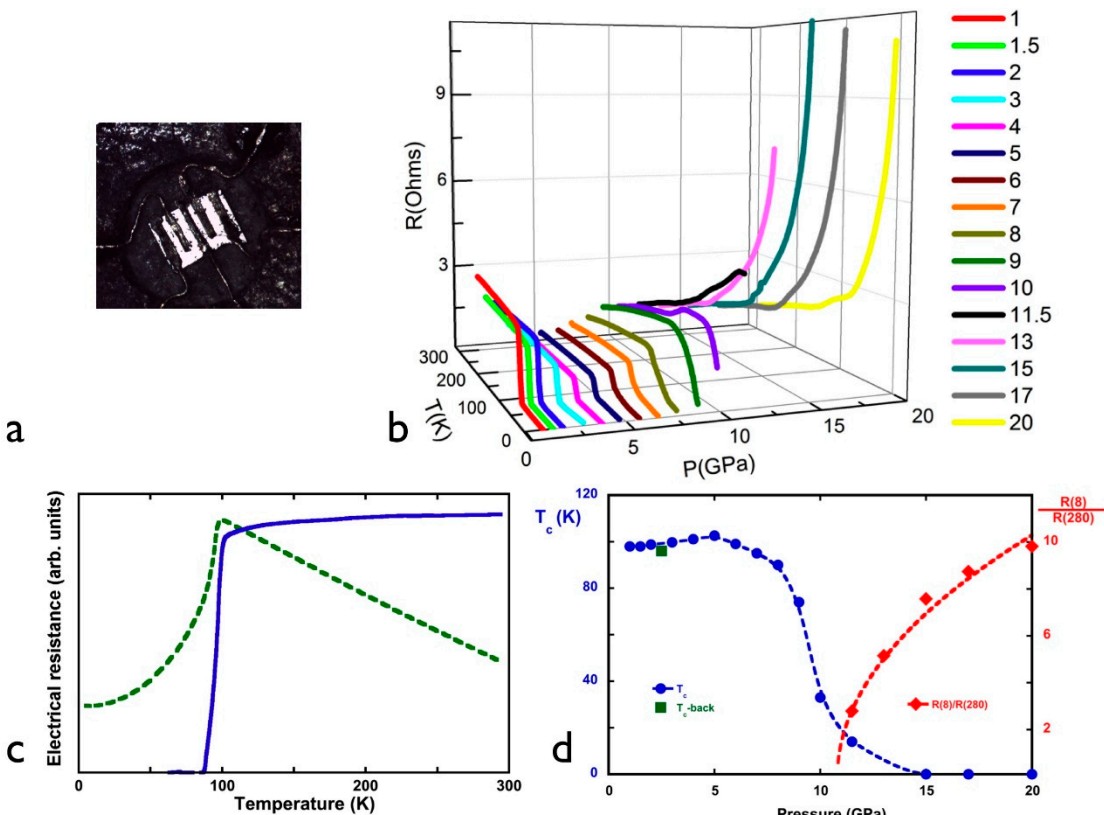

**Figure 1.** (**a**) Single crystal of *Hg-1201* mounted in the pressure cell; (**b**) electrical resistance of the *Hg-1201* single crystal as a function of pressure and temperature. The evolution from a metallic and superconducting behavior to an insulating (indicating charge-order) and non-superconducting one at high pressures is evident; (**c**) comparison of the resistivities at 4 GPa on compression and decompression. The superconducting behavior is recovered although the decompressed sample does not show a clear metallic behavior; and (**d**) in blue circles, we show the evolution of $T_c$ with pressure. It increases up to about 5 GPa and further decreases to attain zero at 15 GPa. The red diamonds correspond to the ratio R(8K)/R(280), a way of showing the increase in the charge ordering with pressure. The dashed red curve is a mean-field power law fit.

In Figure 1d, we show the evolution of the resistivity with pressure. Up to 5 GPa, $T_c$ increases with a slope of 1.2 K/GPa, a value slightly lower than previously reported for nominally optimal doped samples [19,20]. This indicates that the investigated samples are probably well on the summit of the doping parabola. Above 5 GPa, $T_c$ starts decreasing, monotonically reaching a value of zero at 15 GPa. The $T_c$ on decompression is also shown, and almost coincides with the one obtained upon compression. We could not determine a transition temperature towards an ordering that would explain the activated behavior of the resistance, probably due to pressure inhomogeneities, as is often the case in this type of cell. However, we can quantify the passage to an insulating state by plotting the ratio of the low-temperature resistance to the ambient temperature resistance. We plot this evolution, and we are able to plot it with a power law mean field expression $[1 - P/P_c]^{0.5}$, with $P_c$ = 11 GPa.

### 3.2. Synchrotron Measurements

In Figure 2, we show the diffraction patterns on another *Hg-1201* measured in a diamond anvil cell at 24.5 KeV at the CRISTAL beamline of synchrotron Soleil. Three different types of diffraction patterns as a function of pressure can be distinguished. Below 7.5 GPa, we see a well-defined tetragonal spot that increases in intensity with pressure. This indicates the very good crystal quality of the measured crystal. Furthermore, the diffuse streaks, already described as corresponding to fluctuating 1D linear oxygen chains, are also visible [7]. The intensity of the streaks increases around the ($\pm 1$, $\pm 1$, L). For pressures higher than 8 GPa, the streaks around the ($\pm 1$, $\pm 1$, L) Bragg spots of the average structure

develop into diffraction spots. In Figure 2, one of the appearing spots is indicated by a white circle. Besides them, other weak diffraction spots also start to appear, as indicated by green circles in Figure 2. As the pressure increases, the spots around the ($\pm 1$, $\pm 1$, L) points become much more intense and elongated, indicating a strong modification of structure of the sample. Above 13 GPa, strong modifications are observed along the (0, K, L) and (H, 0, L) directions, with the appearance of well-defined spots. Their incommensurate periodicity seems to be unrelated to that of the extra spots around the ($\pm 1$, $\pm 1$, L) spots, thus supporting a phase separation scenario.

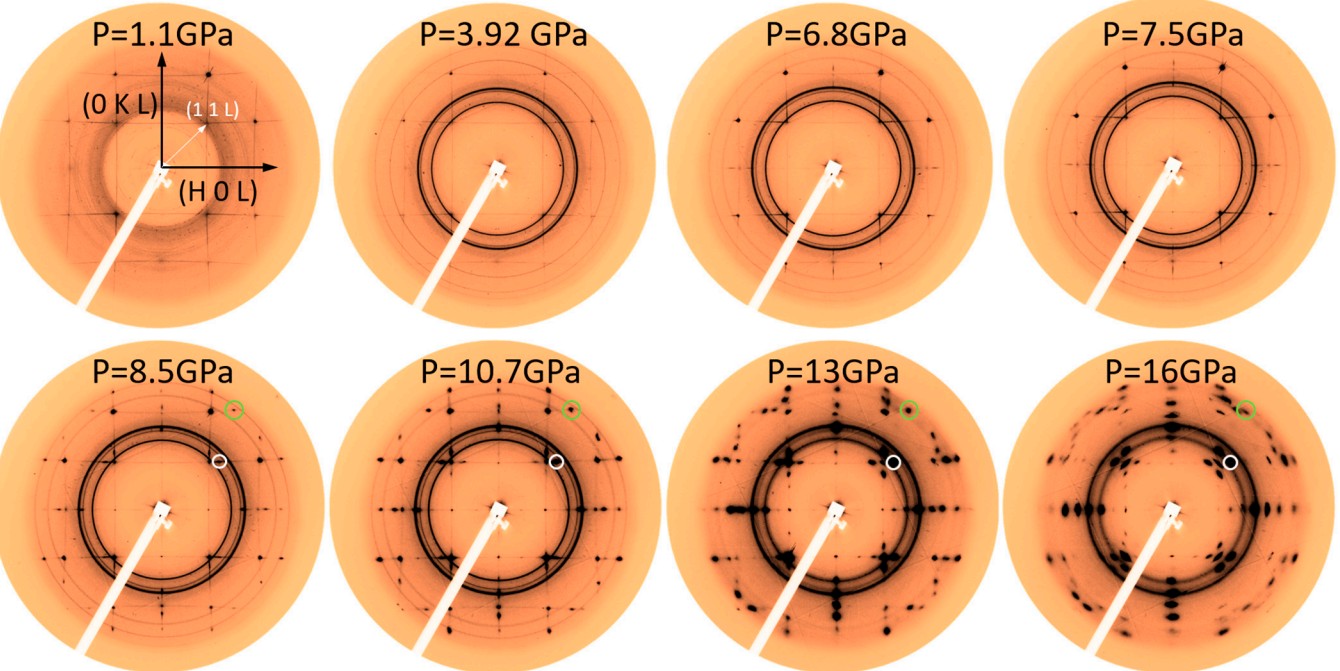

**Figure 2.** Diffraction patterns as a function of pressure. We observe a well-defined tetragonal structure at low pressures, with diffuse stripes that indicate a fluctuating 1D linear oxygen ordering along the two tetragonal axes [7]. Above 8 GPa, we observe the appearance of superstructure spots (indicated by light green and white circles). They become more intense and diffuse at very high pressures.

## 4. Discussion

One can understand the apparition of incommensurable spots by the increase in the correlation between the oxygen chains responsible for the diffuse lines. If this is indeed the case, and since no extra diffuse lines develop in the *a–b* plane, the ordering has to take place along the *c* axis. The indexation of the peaks in the high-pressure range shows that the periodicity corresponds to 8-unit cells along the *c* axis. Correlations along the *c* axis have previously been reported at low pressure on *Hg-1223* cuprates. However, in this case, the reported superstructure had a 5*c* periodicity [21]. Since in our diffraction patterns, incommensurable spots only appear along particular segments of the diffuse lines, this leaves room for other interpretations. One possibility would be the formation of orthorhombic twin domains upon applying pressure. This type of domain has been observed in *YBCO* compounds, giving rise to diffraction spots at variable distance from the tetragonal spots. Another possibility, deriving for the phase separation scenario recently proposed by scanning micro-X-ray diffraction studies, would be that the incommensurate spots have two origins. Some of them would be related to the pressure-induced ordering of the oxygen atoms and the other part to that of the CDW regions. The spots around the ($\pm 1$, $\pm 1$, L) spots have a periodicity in reciprocal space units ($a^* b^* c^*$) of ~($0.25a^*$ $0b^*$ $0.125c^*$), which is close to the ($0.23a^*$ $0b^*$ $0.16c^*$) periodicity observed for the CDW at ambient pressure [9]. The difference may be explained by pressure-induced

changes. The other spots would correspond to a 3D ordering of the oxygen chain along the *c* direction.

We quantified the evolution of the superstructure order by plotting the intensity of the new peaks normalized to the intensity of the neighboring tetragonal Bragg peaks. The results displayed on the left panel of Figure 3 show their evolution with pressure compared with that of $T_c$. This proves that the development of the 3D oxygen ordering destroys the superconducting state by generating a charge order that explains the insulating state observed at high pressure in our resistance measurements.

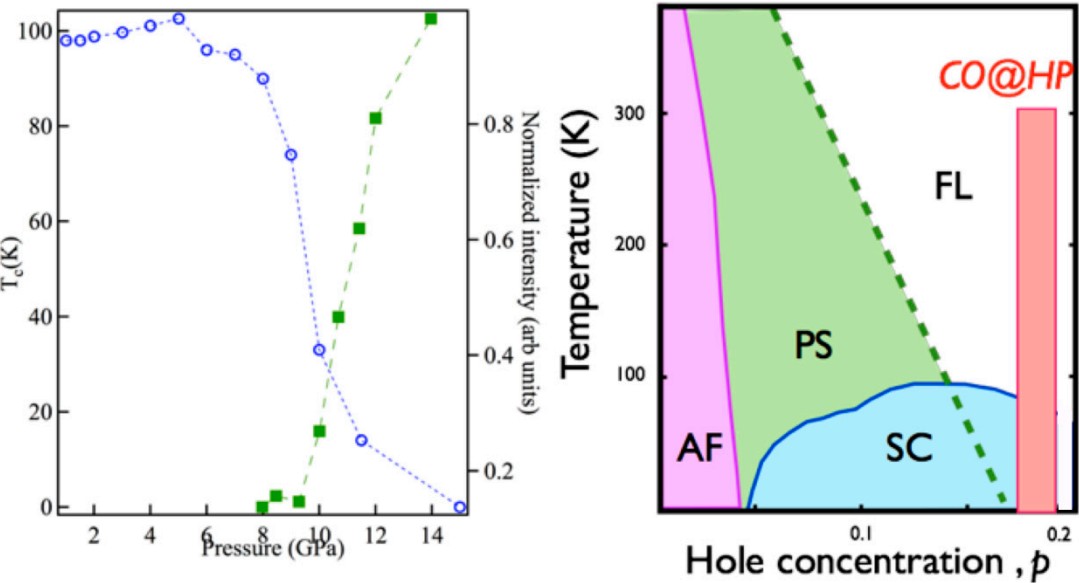

**Figure 3.** (**left panel**) Normalized intensity of the superstructure spots as a function of pressure compared to the evolution of $T_c$; and (**right panel**) phase diagram of mercury cuprates showing where our results are situated as a function of doping.

On the other hand, we can estimate [10] the additional doping introduced the application of 10 GPa to be between approximately 0.02, i.e., dn/dP~0.002 h/GPa. As we started with an optimally doped mono crystal, at *p* = 0.18, we are clearly in the overdoped region. Thus, in strong contrast to almost all previous reports of charge ordering strongly competing with superconductivity in cuprates, only observed in the pseudogap underdoped region, we observed it in the overdoped Fermi-liquid region. Our results therefore support the need to revisit the origin of charge ordering in cuprates reopened after CO in the overdoped region of *Pb–Bi2201* [22].

## 5. Conclusions

Our measurements pose the following fundamental question: as it is observed in a region where the compounds should be in a "normal" Fermi-liquid state, does charge ordering have something to do with the mechanism of high-temperature superconductivity or is it just a phenomenon related to the layered structure of cuprates that has nothing to do with the mechanism of superconductivity? In particular, it was proposed long ago that the low-temperature phases of the $La_{2-x}Ba_xCuO_4$ system are the result of different thermal contractions of the $CuO_2$ and $LaO$ layers [23]. This mechanism would be unrelated to the mechanism of high-temperature superconductivity, and may appear at different concentrations. In this line of thought, one can even wonder whether the mysterious pseudogap region is only the result of the coupling of the interaction responsible for the high $T_c$ with the structural phenomenon and is not an intrinsic property for high-temperature superconductivity. On the other hand, it can be part of the complexity necessary for the appearance of high-temperature superconductivity [24]. Furthermore, it is known that a coherence transition temperature $T_{coh}$ is expected in the overdoped region, with a power law dependence starting from a putative quantum critical point (QCP) whose

exact position on the concentration axis is still a matter of controversy. Our observations could be in a symmetric region of the $p = 0.125$ magical number with respect to the QCP, which remains to be understood. The pressure-induced destruction of filamentary superconductivity in analogy to experiments under strong magnetic fields [25] would also provide an explanation for our high-pressure results. Furthermore, the relation of our observations with the CDWs observed at lower temperatures [6,9] remains to be confirmed. In this respect, pressure-dependent experiments as a function of the temperature are currently being envisaged.

**Author Contributions:** M.I. conceived and coordinated the project, M.N.-R. coordinated the transport measurement under pressure that were realized and analyzed together with D.C.F., D.C. and A.F. prepared and pre-characterized the samples. M.I., J.-P.I., S.R. and P.F. performed the X-ray experiments under pressure. M.I., G.G. and P.F. analyzed the diffraction data. M.I. and H.R. performed the SQUID measurements. M.N.-R. and M.I. wrote the manuscript with contributions from all the authors. All authors have read and agreed to the published version of the manuscript.

**Funding:** D.C.F. acknowledges support from the Brazilian agencies CAPES and Cnpq.

**Institutional Review Board Statement:** Not applicable.

**Informed Consent Statement:** Not applicable.

**Data Availability Statement:** Data can be made available upon request.

**Acknowledgments:** We thank Stephan Megtert and Robert Comès for their contribution to the project. Moreover, M.I. warmly thanks them for driving him into the fascinating world of diffuse scattering and their mentoring for more than a decade on the subject.

**Conflicts of Interest:** The authors declare no conflict of interest.

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
