# Peer review of "Charge Order and Suppression of Superconductivity in HgBa2CuO4+d at High Pressures"

_condensedmatter, doi:10.3390/condmat6030025_

Round 1

Reviewer 1 Report

The paper titled "Charge Order and Suppression of Superconductivity in 2 HgBa2CuO4 at High Pressures" by
Manuel Izquierdo et al.

At ambient pressure diffraction studies have confirmed  that the oxygen interstitials ordering  manifests as diffuse lines in the two tetragonal directions. The paper reports to perform new experiments stuing the evolution of oxygen interstitial ordering with increasing pressure. The paper reports data collected in 300 microns single crystals showing Tc =95K. Tc increases up to a maximum at 5 GPa and it disappears in the ramge 8<P<13 GPa and in the same pressure range  appear.

The new incommensurate  sharp superlattice spots appearing with pressure are similar to the growing  of ordering of oxygen interstitial ordering observed by Fratini in superoxygenated La2CuO4 (Nature, 466(7308), 841-844. (2010) where ordering can be stimulated by x-ray illumination (Nature materials, 10(10), 733-736  (2011).

The authors could expand  their discussion on the suggested phase separation which is of growing interest (see for example

Kagan, M. Y., Kugel, K. I., & Rakhmanov, A. L.  Electronic phase separation: Recent progress in the old problem. Physics Reports Volume 916, Pages 1-106 (15 June 2021)

For the role of the oxygen interstitials wires the authors could cite the seminal work "The gap amplification at a shape resonance in a superlattice of quantum stripes: A mechanism for high Tc." Solid State Communications 100.3 (1996): 181-186.

Guidini, A., & Perali, A. (2014). Band-edge BCS–BEC crossover in a two-band superconductor: physical properties and detection parameters. Superconductor Science and Technology, 27(12), 124002.

Finally the paper is of high scientific interest and should be published with minor changes

Author Response

We thank the referee for the comments and corrections that increase the timeliness and quality of the manuscript.

We tried to satisfy them in the new revised version of the manuscript.

Reviewer 2 Report

This paper reports the discovery of a pressure induced, probably CDW phase and the simultaneous disappearance of superconductivity in overdoped samples.The paper is very interesting and should be published. However, some points need clarification before publication:
1) The abstract reports a [0.25, 0, L] periodicity, however I could not find this periodicity on the body.  Conversely, the c-axis periodicity of the body is not reported on the abstract.

2) There are some other reports of CDW in overdoped systems which is worth mentioning: Peng et al Nature Materials, volume 17, 697–702 (2018). Zhou et al, https://arxiv.org/abs/2103.06094

3) The discussion of the x-ray data is difficult to follow for a non-specialist. Please indicate the streaks and add some (H, K, L) labels to the spots.

4) The ordering parameter behaviour of the resistivity is quite striking. However, this analysis can miss the fact that superconductivity may have a filamentary nature close to the transition, so the resistivity may be zero in the presence of substantial CDW correlations (S. Caprara et al.  SciPost Phys. 8, 1 (2020); B. Leridon New J. Phys. 22, 073025 (2020)). For example the 10GPa data show clear semiconducting like behaviour so the CDW may start at a lower pressure than what the 8K resistivity suggest (see resistivity data in the above papers for related examples). This indeed seems to be the case from X-ray data. It would be good if the authors comment on the possibility of filamentary superconductivity reconciling the two measurements in view of the above works. 

Author Response

The referee's corrections and comments have been of importance in rendering the manuscript more clear and complete.

We included them in the rvised version